# CovDif, a Tool to Visualize the Conservation between SARS-CoV-2 Genomes and Variants

**DOI:** 10.3390/v14030561

**Published:** 2022-03-09

**Authors:** Luis F. Cedeño-Pérez, Laura Gómez-Romero

**Affiliations:** 1Licenciatura de Ciencias Genómicas, Universidad Nacional Autónoma de México, Querétaro 76230, Mexico; pipecedeno@gmail.com; 2Departamento de Bioinformática, Instituto Nacional de Medicina Genómica, Periférico Sur 4809, Arenal Tepepan, Tlalpan, Mexico City 14610, Mexico

**Keywords:** genomic conservation, SARS-CoV-2, conservation landscape, visualization tool, virus genomics, variants of concern, mutation, molecular epidemiology

## Abstract

The spread of the newly emerged severe acute respiratory syndrome (SARS-CoV-2) virus has led to more than 430 million confirmed cases, including more than 5.9 million deaths, reported worldwide as of 24 February 2022. Conservation of viral genomes is important for pathogen identification and diagnosis, therapeutics development and epidemiological surveillance to detect the emergence of new viral variants. An intense surveillance of virus variants has led to the identification of Variants of Interest and Variants of Concern. Although these classifications dynamically change as the pandemic evolves, they have been useful to guide public health efforts on containment and mitigation. In this work, we present CovDif, a tool to detect conserved regions between groups of viral genomes. CovDif creates a conservation landscape for each group of genomes of interest and a differential landscape able to highlight differences in the conservation level between groups. CovDif is able to identify loss in conservation due to point mutations, deletions, inversions and chromosomal rearrangements. In this work, we applied CovDif to SARS-CoV-2 clades (G, GH, GR, GV, L, O, S and G) and variants. We identified all regions for any defining SNPs. We also applied CovDif to a group of population genomes and evaluated the conservation of primer regions for current SARS-CoV-2 detection and diagnostic protocols. We found that some of these protocols should be applied with caution as few of the primer-template regions are no longer conserved in some SARS-CoV-2 variants. We conclude that CovDif is a tool that could be widely applied to study the conservation of any group of viral genomes as long as whole genomes exist.

## 1. Introduction

The newly emerged severe acute respiratory syndrome coronavirus 2 (SARS-CoV-2) is the etiological agent of the Coronavirus Disease 2019 (COVID-19). SARS-CoV-2 is a single-stranded RNA beta-coronavirus with a genome of around 30 thousand base pairs.

Viral genomes are characterized by their small genome size and have the widest variation in mutation rates compared to other biological systems, as their mutation rates can go from 10^−8^ to 10^−4^ substitutions per nucleotide per cell infection (s/n/c). In the case of RNA viruses, the mutation rate goes from 10^−6^ to 10^−4^ s/n/c [1]. SARS-CoV-2 has a proofreading mechanism via the exonuclease ExoN, which gives the SARS-CoV-2 a lower mutation rate compared to other RNA viruses like Influenza virus (10^−6^ and 3 × 10^−5^ per site per cycle, respectively) [2].

The first genome sequence of the virus was released at the beginning of 2020 [3] and since then many genomes have been sequenced and made available in different repositories as the National Center for Biotechnology Information database (NCBI) (www.ncbi.nlm.nih.gov, accessed on 2 March 2022), the Global Initiative for Sharing All Influenza Data (GISAID) database (https://www.gisaid.org, accessed on 2 March 2022), among others, offering a valuable resource for many types of analysis. This information has been used to identify SARS-CoV-2 phylogenetic clades. GISAID clusters SARS-CoV-2 genomes into seven different clades: L being the group where the reference genome is S, V, G, GH, GR and GV, and an additional clade called O is designated for the genomes that do not belong to any of the seven major clades [4]. Also, variants of interest have been identified and analyzed to understand SARS-CoV-2 origin, dispersal and pathogenicity. Some variants display higher transmission rates (for example, b.1.351 has an increase of ~50% [5], and some others display higher mortality rates [6]).

Identifying conserved regions across viral genomes is necessary for many different applications. Conservation is important during the primer design for a PCR test since mutations in the primer regions can affect the sensitivity of a primer set [7,8,9,10]. Conservation is also analyzed during vaccine development, where a conserved region of the virus can help the vaccine became more effective against the virus by giving the host a higher immunity [11]. Identifying these regions can be one of the biggest challenges for highly variable viruses, such as HIV [12].

Alignment software such as Nucmer [13], GSAlign [14], Mauve [15], SibeliaZ [16] or ViralMSA [17] can be used to identify all point mutations, insertions and deletions between any pair of closely related genomes. This information can be processed and the conservation per region can be calculated. However, these software are not designed to produce a genome-wide conservation index per group of genomes and no comparison between groups is provided. Web resources such as Outbreak.info explicitly report and compare mutation prevalence across SARS-CoV-2 lineages [18]. However, most MSA software cannot be run in parallel to handle the processing of hundreds or even thousands of genomes.

Other purpose-specific tools, like GolayMetaMiner, are better designed and optimized for specific purposes. GolayMetaMiner was created to guide the design of RT-PCR primers. GolayMetaMiner computes uniqueness and conservedness scores for each subsequence of a reference genome. The uniqueness score represents if each reference subsequence is only found at the intended target genome and not found in a pool of non-target genomes. The conservedness score represents how conserved each reference subsequence is across a pool of secondary targets. However, GolayMetaMiner is not designed to handle more than two groups of genomes of interest (non-target genomes and secondary targets) nor to compute the differences between all groups of interest. Besides, GolayMetaMiner output is not designed to be compatible with a genome browser like IGV [19].

In this article we present CovDif, a tool that enables the fast visualization of genome-wide conservation across a group of genomes or the difference in conservation between groups, which can be useful to guide the design of detection tests or even as an epidemiological surveillance tool. CovDif generates a conservation landscape for each group of genomes. The conservation landscape indicates the conservation level of all possible reference subsequences across the genomes of interest, i.e., the fraction of the analyzed genomes that are equal to the reference. CovDif does not require the genomes to be closely related. CovDif output can be directly imported into IGV, a genome browser in which any annotation or informative data can be loaded as a separate track. Moreover, CovDif is run in parallel processing hundreds of genomes in seconds. Understanding the conservation across viral genomes is important for the identification of pathogens via sequencing, for the design of detection tests, for the understanding of pathogenicity or even for vaccine development.

## 2. Materials and Methods

### 2.1. Data Collection

SARS-CoV-2 reference genome was downloaded from NCBI (NC_045512.2). It corresponds to the first viral genome sequenced from Wuhan, China. SARS-CoV-2 population genomes were downloaded from the GISAID database. Only complete, high coverage genomes were used. Low coverage genomes were explicitly excluded. Two classes of genomes were downloaded: clade-specific genomes (G, GH, GR, GV, L, O, S and V, sequences submitted between December 2020 and January 2021, nomenclature based on GISAID clades) and variant-specific genomes (b.1.1.7, alpha; b.1.351, beta; p.1, gamma; b.1.525, eta; b.1.427, epsilon; and b.1.1.529, omicron, according to Pango lineages and WHO labels, respectively; sequences submitted before February 2022). Ten thousand genomes per subclass were downloaded when possible. The number of downloaded sequences per subclass is shown in Appendix A as well as the dates of submission. An acknowledgement table of submitting labs for SARS-CoV-2 genomes can be found at Appendix A.

Bacteria and viruses that can be commonly found in the human respiratory tract were obtained from previous studies [20,21,22]. All available sequences from these viruses and bacteria were downloaded from the European Nucleotide Archive (ENA). In addition, genomic sequences from other human coronavirus strains (human coronavirus 229E, human coronavirus OC43 and human coronavirus NL63) were also downloaded from ENA. A total of 1918 environmental sequences were included. Accessions for all sequences can be found at Appendix A.

### 2.2. Construction of the Conservation Landscape

This process can be divided into four sequential stages: definition of reference kmers, alignment, filtering of valid alignments and final summary. To define the reference kmers, the reference genome (NC_045512.2) was split into fragments of size k (from k = 20 to k = 25), with a sliding window of 1, e.g., the first fragment starts at position 1, the second fragment starts at position 2, and so on. Each of these fragments was considered a reference kmer. A conservation landscape was generated for each size k.

Next, the reference kmers were aligned to the genomes of each target group using Razers3 (version 3.5.8, parameters: -ng -mN -i 95) [23] so a conservation landscape was generated for each group of target genomes. To perform the alignment, any nucleotide different to either A, T, C, G or N was changed to N in any genome sequence. To obtain the relaxed conservation landscape only end-to-end identical alignments were considered as valid and no further processing was done. Alternatively, to compute the conservative conservation landscape, alignments including N’s were also considered as valid, and an imputation step was performed: if a region with no valid alignments was surrounded by alignments containing N’s, all the absent alignments for the intermediate kmers were considered as present and valid. This imputation step was required since short alignments in a region with a lot of N’s are usually problematic. Next, the number of genomes aligned against each reference kmer was counted. These counts were divided by the total number of genomes in the group to obtain the frequency of occurrence (a number between 0 and 1) of each reference kmer in the genomes of the group.

Each conservation landscape was saved as a wig file for interoperability. The first column contained the start position for each reference kmer (0-based) and the second column contained the frequency of occurrence which can be interpreted as a metric of conservation.

### 2.3. Construction of the Differential Landscape

Two classes of differential landscapes were built: (i) when only two groups of genomes were analyzed, e.g., A and B and (ii) when more than two groups of genomes were analyzed. In the first class, the differential landscape was obtained by subtracting the frequency of occurrence of any reference kmer at group B of genomes to the frequency of occurrence of the same reference kmer at group A of genomes. In the second class, the differential landscape at any reference kmer was obtained by calculating the absolute value of the difference between the maximum frequency of occurrence and the minimum frequency of occurrence between all the groups. Each differential landscape is saved as a wig file for interoperability. In this case, the second column contained the differential of occurrence which can be interpreted as a metric of differences in conservation.

### 2.4. Genome Browser Interface and Analysis of Conservation and Differential Landscapes

For each analysis all wig files were uploaded to IGV for visualization [24]. Ad hoc python scripts were used to obtain the positions of the drop regions and to obtain the mean value of conservation for the regions of interest.

### 2.5. CovDif Implementation

CovDif is a pipeline developed in Python3 and Bash. It was designed in a modular fashion. The first module computes all conservation landscapes per each group of genomes and the corresponding differential landscape, and the second module loads all generated tracks into the Genome Browser interface. CovDif uses parallel to improve performance by parallelizing all processes [25]. CovDif can be downloaded from [https://github.com/INMEGEN/CovDif, accessed on 2 March 2022].

### 2.6. Annotation of Known Variants

The variants from each clade were obtained from the clade and lineage nomenclature report from GISAID on 2 March 2021 [26]. The mutations associated with each SARS-CoV-2 variant were obtained from the PANGO Lineages Latest Report last accessed 19 August 2021 [27]. These mutations were considered as defining SNPs.

### 2.7. Obtainment of Primer and Probe Sequences from Diagnostic Kits

The Department of health and human services primers and the LKS Faculty of Medicine School of Public Health primers were obtained from WHO documents [28,29] and the primers from BioBasic protocol were obtained from the BioBasic web page [30]. Additional sets of primers were compiled from the literature [31,32,33]. All primer sequences can be found in Appendix A. Each primer (out of 16 different primer pairs) was aligned against the reference genome using BLAST [34].

## 3. Results

### 3.1. Genomic Variation Is Highlighted along the Conservation and Differential Landscapes

CovDif is a tool to obtain and visualize the genetic differences between a reference genome and either one or several groups of target genomes and also across several groups of target genomes. CovDif is useful to analyze the genomic conservation at base pair resolution. CovDif generates one conservation landscape per each group of genomes and one global differential landscape. CovDif is based on the identical alignments of reference kmers (i.e., reference strings of size k).

The conservation landscape is a representation of the number of target genomes that contain each reference kmer along the reference genome. Regions composed by kmers that are present in most target genomes are visualized as horizontal steady lines at high frequency values. In contrast, a single nucleotide variant (SNV) present in a target genome is translated into the absence of all kmers containing the SNV position in the corresponding genome. Consequently, an SNV that has been fixed in the set of target genomes generates k kmers with zero frequency which is visualized as a horizontal steady line at zero in the conservation landscape (Figure 1).

CovDif generates a conservative conservation landscape in which it considers as a match all genomes with either an exact match to the reference genome or a non-perfect match with all mismatches being N’s. This process implies that all N’s are considered as reference bases producing the most conservative estimate about the frequency of the mutant allele. CovDif is also suited to generating a relaxed conservation landscape in which only exact matches are counted, and all N’s are considered as mutant alleles producing a relaxed estimate about the frequency of the mutant allele.

Depending on the application, one could be interested in either the conservative or relaxed conservation landscapes. In all applications discussed in this work, we focused on the conservative landscapes as we were especially interested in the fraction of mutant genomes, unless otherwise stated.

The differential landscape represents the maximum difference in frequency per reference kmer between all genomes being compared. It ranges from −1 to 1 when only two groups are being compared: 1 means that the specific kmer is found in all genomes from group 1 and no genomes from group 2 and −1 implies the same behavior but in the opposite direction. This representation is highly versatile as it can be used to pinpoint either specific, conserved or variable regions. In contrast, this representation ranges from 0 to 1 when more than two groups are being compared. The value at each reference kmer represents the maximum absolute value of all paired differences, where 0 means that the frequency of occurrence of the reference kmer is equal across groups and 1 means that the frequency of occurrence in at least one group is 0 and 1 in another group. In this summarized representation one can readily identify regions that present variation across any of the groups being analyzed (Figure 2).

CovDif does not require the genomes to be from closely related species as it does not require the consecutive reference kmers to be consecutive at the target genomes. CovDif does not align the whole genomes and it analyzes one kmer at a time. These characteristics are useful when comparing distant species or genomes of very different sizes. CovDif does not report the specific nature of the variation even at regions where the reference kmer is not present in any target genome, i.e., a variant has been fixed in the target population; further processing would be required to identify if the variation corresponds to a transition, a transversion, an insertion, a deletion or even several nearby SNVs.

### 3.2. CovDif Can Be Used for a Wide-Range of Applications

#### 3.2.1. It Can Be Used to Identify Species-Specific and Highly Conserved Regions

In some situations, it can be important to determine a set of genomic regions that are present in most population samples from a specific species but not present in either phylogenetically close or distant species that could interfere with the analysis, e.g., in the process of primer design to detect a specific pathogen such as the virus SARS-CoV-2.

We applied CovDif to study the SARS-CoV-2 conservation from population samples and the existence of shared regions with environmental sequences. Two different groups of target sequences were used: (1) sequences from common respiratory viruses and bacteria of the respiratory tract (environmental, possible-confounding sequences), including other human coronaviruses and (2) SARS-CoV-2 genomes obtained from population samples (SARS-CoV-2 genomes). Both of these groups of genomes were compared against the reference SARS-CoV-2 genome aiming to obtain regions not present in group 1 (specific for SARS-CoV-2) and highly-conserved in group 2 (SARS-CoV-2 population samples). Group 1 contains thousands of sequences of very different lengths that belong to very distant organisms and group 2 contains thousands of viral genomes demonstrating the applicability of CovDif to study very distant species and when thousands of viral genomes are analyzed at the same time. For this analysis, the results obtained with a kmer size of 20 and 25 are discussed. The conservation and differential landscapes for genomic positions 19,000 to 25,000 are shown in Figure 3.

In the case of the environmental sequences we observe a contrasting behavior between the relaxed and conservative conservation landscapes. Forty-seven kmers for the relaxed conservation landscape and 5842 kmers for the conservative conservation landscape (out of 29,884 reference kmers for size 20) show a frequency higher than zero (0.15% and 19.54%, respectively). The huge difference between the landscapes indicates that the identity between SARS-CoV-2 kmers and environmental genomes is due mainly to the presence of N’s at the environmental sequences. Moreover, in any case all kmers present in environmental sequences are present in a very low fraction of them. In the case of the relaxed conservation landscape all 47 kmers show a frequency lower than 0.05 and 28 kmers show a frequency lower than 0.01 which means that they are present in less than 5% and 1%, respectively of the environmental sequences. In the case of the conservative conservation landscape all 5842 kmers show a frequency lower than 0.05 and 5823 kmers show a frequency lower than 0.01.

To the contrary, in the case of the conservation landscapes for SARS-CoV-2 population genomes 25,189 and 25,420 out of 26,244 kmers (84.28% and 87.81%, kmer size = 20) in the relaxed and conservative conservation landscapes have a frequency higher than 0.99, meaning that they are conserved in more than 99% of the target genomes. The number of highly conserved kmers slightly decreases as the kmer size increases, reaching 80.11% and 84.17%, respectively at a kmer size equal to 25. This behavior is explained due to the larger number of kmers that overlap with any given mutation as the kmer size increases. Although most kmers are present in a very high proportion of SARS-CoV-2 genomes, no kmer displays a frequency of 1 suggesting that all positions from the SARS-CoV-2 genomes present a variant in at least one target genome. This result should be taken with caution as singletons could be produced by sequencing errors.

A genomic variant generates a drop in the kmer frequency at the conservation landscape. Moreover, the length of the drop is indicative of the specific nature of the variation. An SNV will cause all the kmers overlapping with such a position to no longer exist in the target genome. Nearby SNVs and insertions will produce a longer region with absent kmers. We analyzed how many drop regions were found at the SARS-CoV-2 population genomes using the conservative conservation landscape. A drop region is defined as a set of consecutive kmers (at least k) that present a frequency lower than a fixed threshold surrounded by kmers with frequencies higher than the same fixed threshold. We set a fixed threshold equal to 0.99. We found 127 drop regions: 110 regions presenting frequencies between 0.9 and 0.99, 14 regions in the range between 0.8 and 0.9 and only 3 regions lower than 0.8 displaying frequencies around 0.3. Drop regions with values lower than 0.9 are shown in Appendix A. Two of these regions are localized inside the gene ORF1ab and the other is localized inside the gene S. These results suggest that the SARS-CoV-2 genome is highly conserved at the population level with very few exceptions.

#### 3.2.2. It Can Be Used to Identify Variable Regions between Phylogenetic Clades of the Same Species

Previous studies have identified mutations that classify SARS-CoV-2 population samples in different phylogenetic clades [https://www.gisaid.org/references/statements-clarifications/clade-and-lineage-nomenclature-aids-in-genomic-epidemiology-of-active-hcov-19-viruses/, accessed on 2 March 2022]. We studied if CovDif was able to retrieve such known mutations. Every SARS-CoV-2 clade was analyzed as an independent group of genomes. A sample of genomes for each clade was obtained from GISAID. We show the results for a kmer size of 20 (all other sizes showed the same behavior). As an example, the conservation and differential landscapes for region 1 to 3500 are shown in Figure 3.

All clade-associated mutations were analyzed at the conservation and differential landscape of the corresponding lineage. Appendix A shows the mutations associated with any clade along with any position with a frequency lower than 0.9, the genomic coordinates of the reference kmers overlapping each mutation (start position of first reference kmer followed by start position of last reference kmer), the average frequency at the corresponding conservation landscape across all reference kmers overlapping each mutation (for an S clade-associated mutation, the S clade conservation landscape will be used) and the average frequency at the differential landscape.

As expected, all regions (except by the L-associated mutations) display very low frequencies at the corresponding conservation landscapes as each mutation is present in most of the clade genomes. Moreover, some of the listed mutations have a frequency equal to 1 in the corresponding population (average frequency at conservation landscape equal to 0) as the G11083T, NSP6-L37F and mutations at the V-clade landscape which indicates a mutation present in all of the clade genomes. An opposite behavior is seen for any L clade mutation: frequencies are very close to 1 confirming that the SARS-CoV-2 reference genome used in this study belongs to the L clade [35].

The differential landscape shows the absolute value of the biggest difference between all paired comparisons when more than two groups are being compared, so the differential landscape summarizes all changes seen in any clade (Figure 4). It displays frequencies close to one for any position that dramatically varies across the groups. In Appendix A, it can be seen how the differential landscape shows a value close to 1 for almost any listed position irrespective of the clade of origin. This behavior implies that all mutations are highly prevalent in at least one clade and that no mutation is present in high frequency across all the clades. Mutation 185–204 is the only one with an intermediate value at the differential landscape resulting from its medium frequency at the only clade in which it is present (clade GV). In conclusion, CovDif was able to identify previously known clade-associated mutations.

In addition, we also identified drop regions with a fixed threshold equal to 0.9. We found several regions that have not been previously associated with any clade: one drop region exhibits a frequency equal or lower than 0.001 in clades G, GH, GR and GV; six drop regions display a frequency lower than 0.05 only in clade GV and one in clade V. These low conservation values suggest that these mutations are widely propagated, however they have not been described as clade-associated mutations. We also identified three drop-regions with intermediate frequencies (between 15% and 40%) which could belong to transitory mutations. So, CovDif could be used to highlight transitory mutations displaying intermediate frequencies in actively evolving lineages. Besides, we suggest that CovDif could be applied to find clade-associated regions from any virus with sequencing data from population samples from which a phylogeny could be inferred.

#### 3.2.3. It Can Be Used to Identify Variable Regions between Known Lineages of the Same Species

The same analysis was conducted on SARS-CoV-2 genomes annotated as members of specific variants. A sample of genomes was obtained from GISAID for specific variants. Further analyses were based on a kmer size of 20 (all other sizes showed the same behavior). The conservation and differential landscapes for positions 21,000 to 25,000 are shown in Figure 5. Appendix A shows the average frequency values at the corresponding conservation landscape for any lineage-associated mutations.

The average frequency values at the corresponding conservation landscape are close to 0 for most lineage-associated mutations. This behavior indicates a low prevalence of the reference nucleotide and a high prevalence of the mutant nucleotide in the analyzed genomes. However, some mutations show an unexpected behavior displaying an average frequency close to 1 in some cases. The mutations E-I82T and ORF3a-G174C displayed average frequencies of 0.993 and 1, respectively, in variant B.1.525 and P.1, respectively, indicating the existence of the reference base in most virus genomes. Mutations ORF1a-I4205V, ORF1a-S3158T, ORF1b-P976L and ORF1b-P314L in variant B.1.427/B.1.429, mutation S-Q52R in variant B.1.525 and mutations S-E484K and S-N501Y in variant P.1 displayed average frequencies of 0.281, 0.721, 0.721, 0.232 and 0.310, respectively, indicating the existence of reference and mutated genomes in intermediate frequencies suggesting the possibility of revisiting the definition of a clade-associated mutation. The variant b.1.1.529 is associated with 47 lineage-associated mutations, 28 of them show an average frequency value at the corresponding conservation landscape close to 0 as expected. However, 14 mutations display conservation values between 0.2 and 0.8 suggesting an actively evolving lineage.

#### 3.2.4. It Can Be Used as a Surveillance Tool to Monitor Genomic Variability in Primer and Probe Sequences from Diagnostic Kits

Virus and pathogen sequences can present high mutation rates resulting in lineages with either a higher dispersion rate or higher pathogenicity. Epidemiological surveillance is important for its detection and to establish the sensibility of current detection protocols as new pathogen variants appear. qRT-PCR is the most common tool to detect pathogens due to its high sensibility and specificity. This technique is based on a pair of PCR primers and a probe that generate an amplicon and emit a fluorescence signal when the pathogen is detected. This system relies on PCR primers and probe sequence complementarity to the pathogen genome.

CovDif can be used to easily assess the complementarity (measured as a perfect match) between any known sequence and a set of target genomes. Mutations in probe and primer sequences could impact test sensibility, so CovDif can be used to in-silico monitor detection protocols as a new pathogen propagates and mutates. Sensitivity assessment could be implemented after a mutation in a region of interest has been localized by CovDif. As a case study, we investigated how variable the sequences targeted by the current SARS-CoV-2 detection protocols are in each SARS-CoV-2 variant. We looked for regions in which there is a drop in the frequency values across the corresponding conservation landscape (drop-regions, frequency < 95%), indicating the presence of a mutation in at least 5% of the interrogated genomes. The conservation and differential landscapes for positions 28,000 to 29,500 are shown in Figure 6.

The drop-regions for each SARC-Cov-2 variant found at the primer regions are shown in Appendix A. The reference sequence for most primers is still present in most SARS-CoV-2 genomes irrespective of the variant. However, a few exceptions exist. Most b.1.1.7, b.1.351, b.1.427, b.1.525, P.1 and b.1.1.529 genomes present a mutation in the region 28881-28902 bound by the China-CDC-N-forward primer (conservation landscape frequency = 0.03) [36]. A similar situation is described for the China-CDC-N-reverse primer in the b.1.1.7 variant, for the primers Reza-Mollaei-E-forward and Thailand-WH-NIC-N-reverse in the b.1.1.529 variant, and for the primer CDC-2019-nCov-N3-forward in the variant b.1.525. Moreover, other primers present intermediate frequency values in at least one coverage landscape. However, low conservation found by CovDif at any primer binding region does not imply neither absence of amplification nor a reduction in amplification efficiency. Further studies of these detection protocols in samples belonging to the highlighted variants would be necessary to indicate if the detected mutations interfere with the test sensitivity.

### 3.3. CovDif Can Generate a Conservation Landscape from a List of Variants

Multiple sequence alignments have been widely applied to study the conservation between genomes, usually between closely related genomes [37,38,39]. However, these tools are difficult to apply as the number of genomes becomes extremely large since the complexity of the algorithm increases exponentially. Pair-wise whole genome aligners are also an option as they compare each single genome against a reference to enable the identification of mutations. The result of any of these tools can be translated into a list of mutations. CovDif can take either one or multiple sets of lists of mutations and it can generate either one or several conservation landscapes and a differential landscape. All these landscapes can then be imported into a Genome Browser for visualization and analysis.

## 4. Discussion

The study of sequence conservation and variation along the SARS-CoV-2 genome is important to detect functional regions involved in SARS-CoV-2 dispersion and pathogenesis, such as the multi-basic cleavage site at the Spike protein associated with SARS-CoV-2 virulence or the Spike-D614G mutation involved in viral infectivity [40,41]. Genome-wide catalogues of variation have been created from thousands of genomes [42,43]. Also, online resources such as Nextstrain allow fast and real-time tracking of the SARS-CoV-2 genome-wide diversity along with evolutionary relationships and geographical location [44].

SARS-CoV-2 genomic diversity has been widely characterized. Mutations have been found by Multiple Sequence Aligners (MSA) such as CLUSTAL omega, CLUSTALW and MUSCLE [37,38,39]. From the MSA results custom scripts have been applied to infer SNP profiles [45,46,47,48]. In addition, Nucmer has been used to align SARS-CoV-2 genomes to the Wuhan reference strain and to identify all mutations along each genome [35]. Dotmats have been applied to visualize frequencies of mutations and to highlight differences per group; in this visualization each genome is represented in a row and a point is drawn per each mutation [35].

van Dorp et al., identified 12,807 SNPs present in a set of 47,745 SARS-CoV-2 assemblies (downloaded on July 2020) with the goal of obtaining homoplasies associated with viral transmissibility [42]. Moreover, a study focusing on identifying recurrent mutations as a possible signal of ongoing adaptation of SARS-CoV-2 to its novel human host reported an average pairwise difference of 9.6 SNPs between any two genomes from a curated database of 7666 public genome assemblies downloaded on April 2020.

CovDif is not a Multiple Sequence Aligner and it does not depend on calculating an MSA to compute conservation across the genome. CovDif runs in minutes for hundreds of SARS-CoV-2 genomes while an MSA is a computationally expensive technique. However, the conservation landscape generated from one group of genomes is similar to the conservation plot that can be derived from an MSA by visualization tools like Jalview [49]. Nevertheless, an MSA is not designed to calculate and visualize conservation differences between groups.

CovDif interrogates genomic conservation at the kmer level identifying species-specific and highly conserved regions. CovDif does not require the genomes to be closely related since it does not perform any whole-genome alignment. CovDif can also identify variable regions between phylogenetic clades and it can identify variable regions between known lineages. Noteworthy, CovDif can identify variable regions between any group of genomes that can be dynamically defined by the end user. Besides, CovDif pinpoints mutations with intermediate frequencies that could be actively evolving in any clade or lineage. However, as CovDif does not perform an MSA, it cannot be used to calculate phylogenetic distances or genomic synteny. CovDif is not a variant caller and it does not identify the specific genotype of the mutant regions. CovDif can also be used to visualize the genomic conservation in any regions of interest, such as regions bound by specific detection and diagnostic protocols. These regions can also be dynamically defined by the end user allowing real-time exploration of genomic conservation. However, CovDif is only an exploratory tool as low conservation found by CovDif at any primer binding region does not imply neither absence of amplification nor a reduction in amplification efficiency

CovDif can output frequency files per genome which allow the identification of reference kmers missing at any specific target genome. These missing kmers are the result of mutations at the target genome. By counting these missed regions we conclude that there are around either 17,600 mutations or 31,586 mutations, if N’s are considered as reference or non-reference bases, respectively, in the 55,696 SARS-CoV-2 genomes. We also conclude that the average difference between any SARS-CoV-2 genome and the reference is 13.56 or 14.39, if N’s are considered as reference or non-reference bases, respectively. These estimates are in concordance with previous estimates about SARS-CoV-2 diversity. One limitation of our approach is that adjacent mutations, closer than 20bp (the kmer length used in this study), could produce one wider missed region in any given genome, resulting in a sub-estimation of the number of mutations for highly variable genomes.

Identification of conserved RNA regions in SARS-CoV-2 is important for the design of antivirals or to guide diagnostics. A previous study identified RNA stretches longer than 15 nucleotides conserved between SARS-CoV-2 and a range of beta coronaviruses. Rangan et al., identified 30 regions, 10 of which are longer than 20 nucleotides, completely conserved in SARS-related complete genome sequences and in around 700 SARS-CoV-2 sequences [50]. For all of these 10 regions we obtained the median of the frequency of all immersed kmers from the conservative conservation landscape built from 55,696 SARS-CoV-2 genomes. We observed frequency values larger than 0.95 for all the regions and frequency values larger than 0.99 in 8 of them. For all of them we observed a frequency equal to 0 in the case of the environmental landscape. Interestingly, we could not find the presence of an exact match for any of these regions in SARS-related sequences such as SARS-Cov (SARS coronavirus HSR 1, accession AY323977.2) and MERS-Cov (Middle East respiratory syndrome-related coronavirus, accession: NC_019843.3).

The characterization of sequence conservation between SARS-CoV-2 and MERS-CoV as well as other seasonal human coronaviruses such as HKU1, OC43, NL63 and 229E is important for the establishment of RT-PCR primers and probes. Anantharajah et al., reported that no significant homologies were found between these organisms, suggesting a low risk of false positives during the RT-qPCR diagnosis [51]. Our study confirms such findings as the number of shared kmers between SARS-CoV-2 and SARS coronavirus HSR 1, MERS-CoV and the human coronavirus HKU1, OC43, NL63 and 229E is 1628, 1, 18, 8, 0 and 18, respectively out of 29,884, i.e., 5% of the total kmers in the case of SARS coronavirus HSR 1, the most similar virus to SARS-CoV-2.

Mutations that were not described as clade associated but that presented low frequencies at the conservation landscape (i.e., high frequency of the mutant allele in the population) have been previously reported. The mutation at position 14,408 identified at several SARS-CoV-2 clades in this study has been previously reported as a recurrent mutation in Europe [52]. Mutations at positions 445, 6286, 26,801 and 29,645 were first associated with a monophyletic group stemming from the larger 20A clade [53]. Besides, mutation 26,801 has been described as a B.1.1.7 identifying mutation [54]. In another study, they reported an increase in the frequencies of mutations 204 and 27,944 that were found at intermediate frequencies at clade GV in our study [55]. An analysis of the frequency of all mutations identified at clade GV over time revealed different temporal patterns with most variants fluctuating in frequency over time (Appendix A).

RT-qPCR is the most reliable detection protocol to identify SARS-CoV-2. Different factors could impair detection efficiency such as viral load and primer-template mismatches depending on their genomic context and the nature of the substitution [56]. Low viral load could hinder proper pathogen identification. Moreover, a study showed that 40–60% of positive SARS-CoV-2 clinical samples are not detected when they exhibit Ct values lower than 30 [51]. Additionally, a systematic study quantitatively investigated the effect of different mismatches at the 3′-end region of real-time PCR primers. They found that single mismatches could have from minor to severe impact on amplification efficiency. This effect has also been observed during influenza detection by RT-qPCR, in which a single primer-template mismatch reduces the assay sensitivity [57]. Therefore, monitoring of genomic conservation, especially on the regions complementary to the RT-qPCR primers and probes, is critical to ensure proper pathogen identification as any pathogens propagate and mutate and new variants emerge.

## Figures and Tables

**Figure 1 viruses-14-00561-f001:**
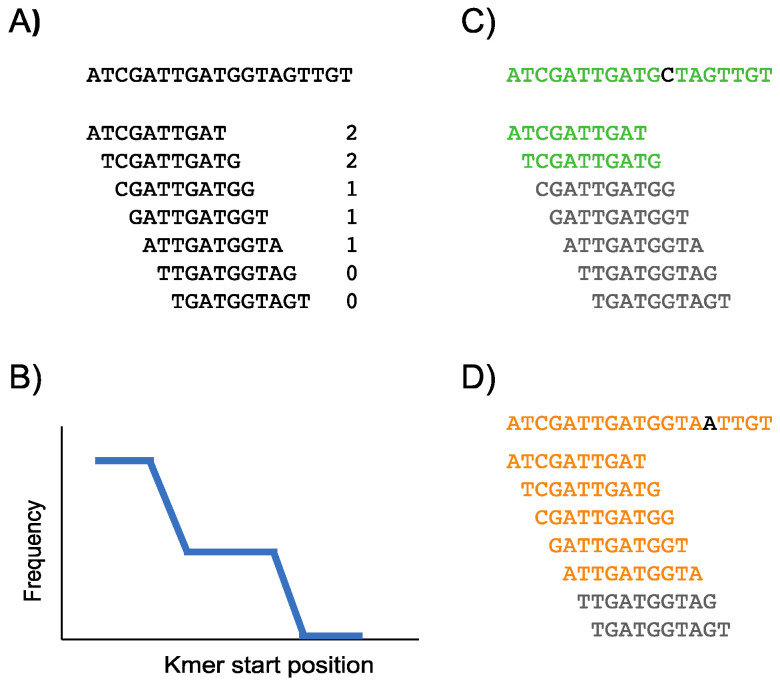
Schematic representation of the construction of a conservation landscape. (**A**) A reference genome and some of its reference kmers are shown. The numbers in front of each kmer are the number of target genomes that contain each kmer. The target genomes that were considered to generate this conservation landscape are shown in (**C**,**D**). (**B**) The graphical representation of the information presented in (**A**). (**C**) A target genome is shown in green. This genome has a G to C mutation that is shown in black. Each reference kmer is looked for in the target genome. The found reference kmers are shown in green and the missing reference kmers are shown in gray. (**D**) Same information is shown for a different target genome depicted in orange. This genome has a G to A mutation.

**Figure 2 viruses-14-00561-f002:**
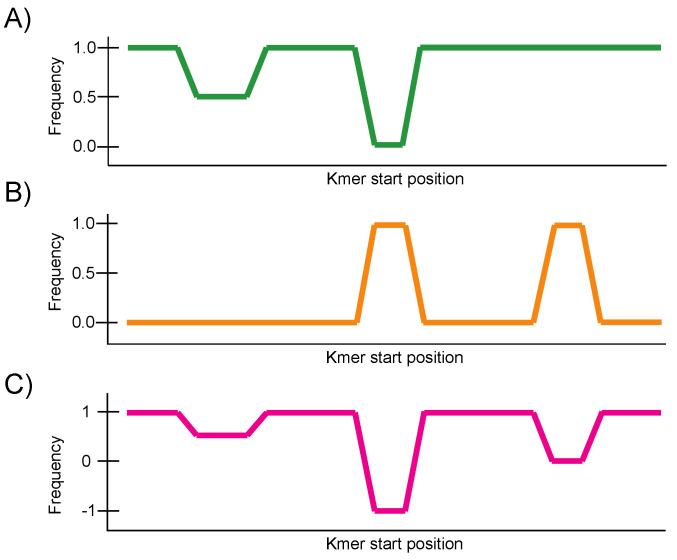
Schematic representation of a differential landscape between two groups. (**A**) The conservation landscape for group 1 target genomes. (**B**) The conservation landscape for group 2 target genomes. (**C**) The differential landscape between group 1 and 2. The differential landscape is obtained by subtracting conservation at group 2 from conservation at group 1. The differential landscape goes to 1 when conservation is 1 in group 1 and 0 in group 2. In contrast, the differential landscape goes to −1 when conservation is 0 in group 1 and 1 in group 2.

**Figure 3 viruses-14-00561-f003:**
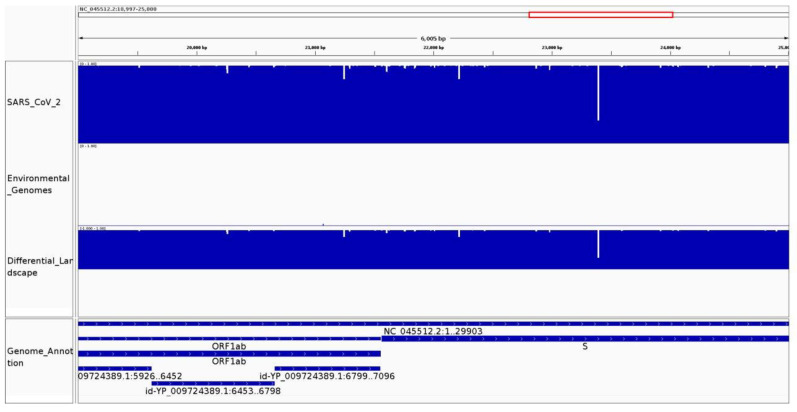
Conservation and differential landscapes for SARS-CoV-2 population genomes and environmental genomes. The visualization generated by CovDif via IGV for the genomic interval 19,000–25,000 is shown. Top track: conservation landscape for SARS-CoV-2 population genomes, middle track: conservation landscape for environmental genomes and bottom track: differential landscape showing the difference between the two groups.

**Figure 4 viruses-14-00561-f004:**
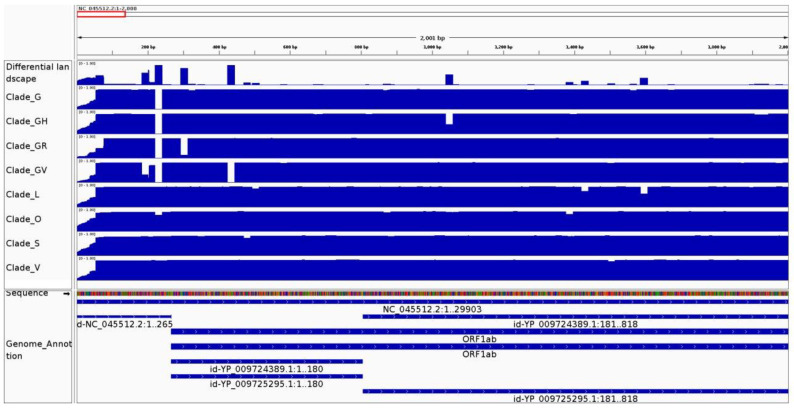
Conservation and differential landscapes for each SARS-CoV-2 clade. The visualization generated by CovDif via IGV for the genomic interval 1–3500 is shown. Top track: differential landscape showing the biggest difference between the all groups; middle tracks: conservation landscape for each SARS-CoV-2 clade (according to GISAID) and bottom track: SARS-CoV-2 annotation.

**Figure 5 viruses-14-00561-f005:**
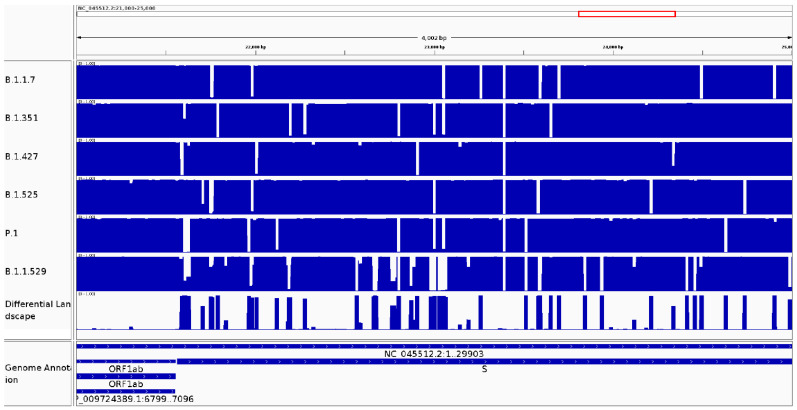
Conservation and differential landscapes for SARS-CoV-2 variants. The visualization generated by CovDif via IGV for the genomic interval 21,000–25,000 is shown. Top tracks: conservation landscape for each SARS-CoV-2 variant; middle track: differential landscape showing the biggest difference between all groups; and bottom track: SARS-CoV-2 annotation.

**Figure 6 viruses-14-00561-f006:**
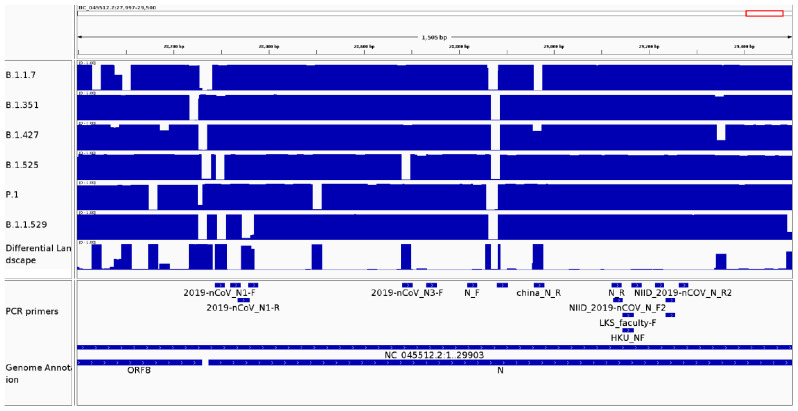
Conservation and differential landscapes for a SARS-CoV-2 region with low conservation in a region targeted by a PCR primer. The visualization generated by CovDif via IGV for the genomic interval 28,000 to 29,500 is shown. Top track: differential landscape showing the biggest difference between all groups; middle tracks: conservation landscape for each SARS-CoV-2 variant; last to bottom track: shows the regions targeted by the PCR primers; and bottom track: SARS-CoV-2 annotation.

## Data Availability

SARS-CoV-2 sequences were obtained from GISAID. The accession numbers for these sequences can be found at https://github.com/INMEGEN/CovDif, accessed on 2 March 2022. Environmental sequences were downloaded from ENA. The accession numbers for these sequences can be found at https://github.com/INMEGEN/CovDif, accessed on 2 March 2022.

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
