# Peer review of "CovDif, a Tool to Visualize the Conservation between SARS-CoV-2 Genomes and Variants"

_viruses, 2022, doi:10.3390/v14030561_

Round 1

Reviewer 1 Report

Major Points:
- Introduction should comment on other similar tools for conservation analysis and visualization of conservation. These tools should be contrasted against this tool for shortcomings, advances, etc.
- Supplemental materials should include an acknowledgement table of submitting labs for GISAID genomes per GISAID's data usage guidelines - see an example on their site
- For reproducibility, the accessions for all genomes or other sequences should be included as a supplemental table. This is specifically needed for methods section lines 77-82
- For Figure 1, using a fixed width font for nucleotide sequences will make the comparisons more clear. Also, when listing the k-mers for the purple and orange genomes, the SNV should be included in the k-mer. It would make it more clear as to why those sequences are different to the reference and therefore colored grey. 
- Figure 1b is confusing. Green seems like it should go at the bottom of the figure to keep the story consistent with the single genome comparison. Also, why does orange range from 0 to 1? Per description on line 179 since it's a reference vs single genome it should be from -1 to 1. The accompanying description is also confusing. This could benefit from a schematic integrated to figure 1 or describing criteria more clearly in a list form. 
- In the SARS-CoV-2 vs environmental comparison (Figure 2 + line 229-231) this comparison could potentially be improved by using a low complexity masking tool like Dustmasker on the reference genome before computing k-mers and likewise on environmental genomes. Then the low information content will not be highlight when the differences are calculated and you'd instead see true genomic differences in conservation.
- The figures from the visualization could be improved for clarity. Also, the location labels and track labels are clearly overlaid on the screenshots of the visualization. So the visualization itself has been manipulated beyond what the tool is providing. This could also be masking or misleading for other aspects of the visualization. If the paper is around a viz tool, those images should be directly from the tool. If resolution is too low, consider including a video as a supplemental file where you can zoom in. Regarding mutations and conservation differences consider leveraging color or other features to make the contrasts more clear.

Minor Points:
- In abstract, include at time of review or proofing for case counts and deaths or as of <insert applicable date>
- line 15, pandemic evolves
- line 17, include which type of genomes/scope of comparisons? All SARS-CoV-2, all Coronaviridae, etc.
- line 19, what is meant by conservation differences? are these point mutations, mutation regions, etc. be more specific
- line 19, to which clades - all? within a time period? the most predominant?
- line 20, inconsistent capitalization of your own tool - unify throughout
- line 20, "defining SNP" what do you mean? variant defining SNP? 
- conservation landscape is not a widely used term and would benefit from being defined early in the context the authors mean for this paper
- line 30, incorrect acronym, virus -> coronavirus 2
- line 33, you don't need the case and death counts again. the world knows the pandemic is harsh and these numbers are incorrect nearly immediately after writing them
- line 49, typo - were -> where
- line 131 missing a space after period, line 133 space between reference and period - clean these kind of things up throughout manuscript
- Case consistency SARS-CoV-2 throughout
- For ease of reading, where possible or at least on first mention include the Greek label as well e.g. B.1.351 (Beta) (example line 53)
- Include versions where applicable of all software used e.g. RazerS 3 line 91 and which parameters were used even if default values
- line 86, split instead of splitted
- "By the other hand" should be "on the other hand" or even better "alternatively" or "in contrast"

Author Response

We thank the reviewer for the comments. We include a point-by-point response. The line numbers correspond to the Word document with track changes:

Major Points:

- Introduction should comment on other similar tools for conservation analysis and visualization of conservation. These tools should be contrasted against this tool for shortcomings, advances, etc.

We commented on other tools that could be used to study conservation on the main manuscript. We discussed their advantages and shortcomings and compared them to CovDif (L75-104)

- Supplemental materials should include an acknowledgement table of submitting labs for GISAID genomes per GISAID's data usage guidelines - see an example on their site

We added all acknowledgement tables on the Supplemental materials as Supplementary_File_1 (L122-123)

- For reproducibility, the accessions for all genomes or other sequences should be included as a supplemental table. This is specifically needed for methods section lines 77-82

We included accessions for all sequences on the Supplemental materials as Supplementary_File_2 (L129-130)

- For Figure 1, using a fixed width font for nucleotide sequences will make the comparisons more clear. Also, when listing the k-mers for the purple and orange genomes, the SNV should be included in the k-mer. It would make it more clear as to why those sequences are different to the reference and therefore colored grey.

We changed the font to Courier which is a monospaced font.

All kmers shown in the image are reference kmers. The ones colored either purple or orange are present at the purple and orange target genomes, respectively. The ones colored gray are not found in the respective target genome due to the presence of a mutation, which is highlighted in black in either the purple or orange target genomes. We split Figure 1 in two figures and we included a more straightforward explanation (L215-223)

- Figure 1b is confusing. Green seems like it should go at the bottom of the figure to keep the story consistent with the single genome comparison. Also, why does orange range from 0 to 1? Per description on line 179 since it's a reference vs single genome it should be from -1 to 1. The accompanying description is also confusing. This could benefit from a schematic integrated to figure 1 or describing criteria more clearly in a list form.

Figure 1B is not designed based on Figure 1A. The goal of Figure 1A is to represent the construction of the conservation landscape and the goal of Figure 1B is to represent the construction of the differential landscape showing the relationship between the values at the conservation landscapes of the groups being compared and the values at the differential landscape. We agreed that this is confusing. So, we split Figure 1 in two figures and included a more straightforward explanation (L268-274)

- In the SARS-CoV-2 vs environmental comparison (Figure 2 + line 229-231) this comparison could potentially be improved by using a low complexity masking tool like Dustmasker on the reference genome before computing k-mers and likewise on environmental genomes. Then the low information content will not be highlight when the differences are calculated and you'd instead see true genomic differences in conservation.

We use Dustmasker on the SARS-CoV-2 reference genome. We found almost no difference between the SARS-CoV-2 masked genome and the SARS-CoV-2 unmasked genome. The only masked bases are 33 consecutive A’s found at the 3’ end of the genome. This implies that the results for the last 13 kmers of the genome would be the only ones that could vary after applying Dustmasker

We also use Dustmasker on each of the environmental genomes. We calculated the relaxed and conservative conservation landscapes based on either the masked or unmasked environmental genomes. Importantly, Razers3 considers all masked bases as Ns. The relaxed conservation landscape will consider all N’s as non reference alleles and the conservative conservation landscape will consider all N’s as reference alleles. We can observe how the relaxed conservation landscape is the same in both cases: for either the masked or unmasked environmental genomes. On the contrary, the conservative conservation landscape is even more noisy when the masked environmental genomes are considered. This is due to the presence of masked bases that will be considered as N’s by Razers3 and as reference alleles by CovDif producing a false impression of conservation.  So, we conclude that the use of masking algorithms will not benefit CovDif

- The figures from the visualization could be improved for clarity. Also, the location labels and track labels are clearly overlaid on the screenshots of the visualization. So the visualization itself has been manipulated beyond what the tool is providing. This could also be masking or misleading for other aspects of the visualization. If the paper is around a viz tool, those images should be directly from the tool. If resolution is too low, consider including a video as a supplemental file where you can zoom in. Regarding mutations and conservation differences consider leveraging color or other features to make the contrasts more clear.

The reviewer is right, we manually added  location labels and track labels to improve visualization. We updated all images to include the image provided by IGV avoiding all manual modifications.

We would like to clarify that our tool generates all files that are loaded into IGV as tracks for visualization. So, the visualization capabilities are restricted by IGV capabilities. To our knowledge, there is no way to add the color contrasts suggested by the reviewer.

Minor Points:

- In abstract, include at time of review or proofing for case counts and deaths or as of <insert applicable date>

done

- line 15, pandemic evolves

done

- line 17, include which type of genomes/scope of comparisons? All SARS-CoV-2, all Coronaviridae, etc.

Viral genomes (L17)

- line 19, what is meant by conservation differences? are these point mutations, mutation regions, etc. be more specific

We specified the type of mutations that will cause loss of conservation (L19-21)

- line 19, to which clades - all? within a time period? the most predominant?

We specified which clades in the abstract and the time period in Material and Methods (L117-118)

- line 20, inconsistent capitalization of your own tool - unify throughout

done

- line 20, "defining SNP" what do you mean? variant defining SNP?

The meaning of this phrase was added to methods [L188]

- conservation landscape is not a widely used term and would benefit from being defined early in the context the authors mean for this paper

We added a definition at the end on the introduction (L95-98)

- line 30, incorrect acronym, virus -> coronavirus 2

done

- line 33, you don't need the case and death counts again. the world knows the pandemic is harsh and these numbers are incorrect nearly immediately after writing them

done

- line 49, typo - were -> where

done

- line 131 missing a space after period, line 133 space between reference and period - clean these kind of things up throughout manuscript

We cleaned this type of errores  throughout manuscript

- Case consistency SARS-CoV-2 throughout

done

- For ease of reading, where possible or at least on first mention include the Greek label as well e.g. B.1.351 (Beta) (example line 53)

done (L119)

- Include versions where applicable of all software used e.g. RazerS 3 line 91 and which parameters were used even if default values

The parameters for Razers3 were added as well as the python version (L139 and L178)

- line 86, split instead of splitted

done

- "By the other hand" should be "on the other hand" or even better "alternatively" or "in contrast"

We verified the use of this conector throughout manuscript

Reviewer 2 Report

The authors need to compare the findings to Development of a Novel, Genome Subtraction-Derived, SARS-CoV-2-Specific COVID-19-nsp2 Real-Time RT-PCR Assay and Its Evaluation Using Clinical Specimensby Yip et al in 2020 in IJMS. Personally I do not find any novelty in the current study and it adds very little to the literature. 

Round 2

Reviewer 2 Report

Thank you for the revised version. Just some minor comments -

  1. The explanation "CovDif does not require the genomes to be closely related as long as they are composed of one single chromosome." seems to contradict the premises of the study. If CovDif is meant to study SARS-CoV-2, then the genomes should be somewhat similar. It would make little sense to compare very divergent genomes or genomes that have no clear homology, e.g. SARS-CoV-2 vs smallpox virus, or SARS-CoV-2 vs E. coli
  2. Multiple sequence alignment for linear genomes and the visualisation of sequence conservation is a problem solved around 20 years ago. For visualisation, the authors could consider Jalview:

Waterhouse AM, Procter JB, Martin DMA, Clamp M, Barton GJ (2009) Jalview Version 2-a multiple sequence alignment editor and analysis workbench. Bioinformatics 25: 1189-1191. doi:10.1093/bioinformatics/btp033

For k-mer based phylogenetic analysis, as the authors have hinted in their "multi-group" comparison (if not for phylogenetic analysis, there would arguably be little point in comparing multiple groups), they may refer to CVTree -

Nucleic Acids Res . 2004 Jul 1;32(Web Server issue):W45-7. doi: 10.1093/nar/gkh362.

Nucleic Acids Research, Volume 37, Issue suppl_2, 1 July 2009, Pages W174–W178, https://doi.org/10.1093/nar/gkp278

Guanghong Zuo, Bailin Hao (2015) CVTree3 web server for whole-genome-based and alignment-free prokaryotic phylogeny and taxonomy, Genomics Proteomics & Bioinformatics, 13: 321-331.

Author Response

We appreciate the reviewer’s comments. We include a point-by-point response

Thank you for the revised version. Just some minor comments -

  1. The explanation "CovDif does not require the genomes to be closely related as long as they are composed of one single chromosome." seems to contradict the premises of the study. If CovDif is meant to study SARS-CoV-2, then the genomes should be somewhat similar. It would make little sense to compare very divergent genomes or genomes that have no clear homology, e.g. SARS-CoV-2 vs smallpox virus, or SARS-CoV-2 vs E. coli

CovDif can be used for a wide range of applications. It can be used to study conservation between SARS-CoV-2 clades or lineages (as shown in section 3.2.2 and 3.2.3 of the manuscript) in which case all analyzed genomes will be closely related. However, CovDif can also be used to determine a set of genomic regions that are present in most  SARS-CoV-2 population samples but not present in phylogenetically distant species like common respiratory viruses and bacteria of the respiratory tract. One potential application for these regions will be to be used as PCR primer for specific amplification of  SARS-CoV-2 (as shown in section 3.2.1 of the manuscript).

  1. Multiple sequence alignment for linear genomes and the visualisation of sequence conservation is a problem solved around 20 years ago. For visualisation, the authors could consider Jalview:

Waterhouse AM, Procter JB, Martin DMA, Clamp M, Barton GJ (2009) Jalview Version 2-a multiple sequence alignment editor and analysis workbench. Bioinformatics 25: 1189-1191. doi:10.1093/bioinformatics/btp033

CovDif is not a Multiple Sequence Aligner and it does not depend on calculating a MSA to show conservation across the genome. CovDif runs in minutes for hundreds of genomes while a MSA is a computationally expensive technique. However, it is true that the conservation landscape generated from one group of genomes is similar to the conservation plot that can be derived from a MSA. Nevertheless, an MSA is not designed to calculate and visualize the conservation differences between groups. We have added this explanation to the discussion section of the manuscript (L534-539)

For k-mer based phylogenetic analysis, as the authors have hinted in their "multi-group" comparison (if not for phylogenetic analysis, there would arguably be little point in comparing multiple groups), they may refer to CVTree -

Nucleic Acids Res . 2004 Jul 1;32(Web Server issue):W45-7. doi: 10.1093/nar/gkh362.

Nucleic Acids Research, Volume 37, Issue suppl_2, 1 July 2009, Pages W174–W178, https://doi.org/10.1093/nar/gkp278

Guanghong Zuo, Bailin Hao (2015) CVTree3 web server for whole-genome-based and alignment-free prokaryotic phylogeny and taxonomy, Genomics Proteomics & Bioinformatics, 13: 321-331.

CovDif goal is not to perform any kind of phylogenetic analysis. CovDif does not compute any distance metric that could be used directly as a pairwise distance between each of the genomes or between each of the groups of genomes. Its purpose is not to establish phylogenetic relationships. For all of these reasons, we believe that CVTree is out of the scope of our work.
